# Selective Isolation and Identification of Microorganisms with Dual Capabilities: Leather Biodegradation and Heavy Metal Resistance for Industrial Applications

**DOI:** 10.3390/microorganisms12051029

**Published:** 2024-05-20

**Authors:** Manuela Bonilla-Espadas, Basilio Zafrilla, Irene Lifante-Martínez, Mónica Camacho, Elena Orgilés-Calpena, Francisca Arán-Aís, Marcelo Bertazzo, María-José Bonete

**Affiliations:** 1INESCOP-Footwear Technological Centre, 03600 Alicante, Spain; mbonilla@inescop.es (M.B.-E.); ilifante@inescop.es (I.L.-M.); eorgiles@inescop.es (E.O.-C.); aran@inescop.es (F.A.-A.); mbertazzo@inescop.es (M.B.); 2Grupo Biotecnología de Extremófilos, Departamento de Bioquímica y Biología Molecular y Edafología y Química Agrícola, Universidad de Alicante, 03690 Alicante, Spain; basilio.zafrilla@ua.es (B.Z.); camacho@ua.es (M.C.)

**Keywords:** heavy metal reduction, leather biodegradation, microbial strain characterisation, tannery wastewater

## Abstract

Tanning, crucial for leather production, relies heavily on chromium yet poses risks due to chromium’s oxidative conversion, leading to significant wastewater and solid waste generation. Physico-chemical methods are typically used for heavy metal removal, but they have drawbacks, prompting interest in eco-friendly biological remediation techniques like biosorption, bioaccumulation, and biotransformation. The EU Directive (2018/850) mandates alternatives to landfilling or incineration for industrial textile waste management, highlighting the importance of environmentally conscious practices for leather products’ end-of-life management, with composting being the most researched and viable option. This study aimed to isolate microorganisms from tannery wastewater and identify those responsible for different types of tanned leather biodegradation. Bacterial shifts during leather biodegradation were observed using a leather biodegradation assay (ISO 20136) with tannery and municipal wastewater as the inoculum. Over 10,000 bacterial species were identified in all analysed samples, with 7 bacterial strains isolated from tannery wastewaters. Identification of bacterial genera like *Acinetobacter*, *Brevundimonas*, and *Mycolicibacterium* provides insights into potential microbial candidates for enhancing leather biodegradability, wastewater treatment, and heavy metal bioremediation in industrial applications.

## 1. Introduction

The leather industry is vital in the global economy, providing high-quality fashion, automotive, and upholstery products [1]. Tanning is the chemical process by which collagen fibres are stabilised, preventing their putrefaction and ultimately forming durable leather [2]. For over 160 years, conventional chrome tanning has been employed in leather manufacturing [3], endowing leather with excellent shrinkage temperature (Ts) [4] and mechanical properties like flexibility, durability, and resistance to environmental factors [5]. Currently, 90% of hides are tanned using chrome [6], where chromium ions (Cr^3+^) crosslink with the carboxyl and amino groups in the collagen fibres [7]. However, chrome tanning agents face significant limitations due to potential hazards, with the most concerning being the oxidative conversion of Cr(III) into hazardous and carcinogenic Cr(VI) [8]. This oxidation process could occur during the tanning process due to elevated pH levels, temperature fluctuations, exposure to UV radiation, improper storage conditions, and the use of lubricants containing double bonds in their molecular structure [9]. At the micro-level, Cr(III) is an essential trace element for multiple physiological processes in the human body, such as glucose, fat, and the protein metabolism, by enhancing insulin activity [10]. While Cr(III) complexes face challenges in penetrating cell membranes [11], Cr(VI) is highly soluble in water and toxic. It can pass rapidly through cell membranes, accumulating and, eventually, interacting with proteins and nucleic acids, ultimately damaging DNA [12]. 

The leather goods industry was worth USD 245 billion in 2022 [13] and stands out as one of the most environmentally impactful and resource-intensive sectors. From every 1000 kg of raw material, 250 kg of leather is produced, leaving a substantial water footprint ranging from 15,000 to 120,000 cubic meters [14]. This process results in the generation of 15 to 50 metric tons of wastewater and 400 to 700 kg of solid waste, greenhouse gases (such as CO_2_, H_2_S, and NH_3_), as well as volatile organic compounds like amines, aldehydes, and hydrocarbons [12]. The emission of chemicals is strongly influenced by the treatment type and the technological processes employed in tanneries [15]. 

Various methods are employed to remove heavy metals from inorganic eluents [16]. Physico-chemical approaches include precipitation through the use of metal hydroxides, sulphides, carbonates, and phosphates [17]. Ion exchange utilizes solid resins for reversible ion exchange [18]. Membrane filtration methods include nanofiltration (NF) for molecules within the 300–500 Da molecular weight range [19] and reverse osmosis (RO) through a pressure-driven separation [20]. Other recent alternative techniques have also been employed, such as coagulation/flocculation, electrocoagulation, electro-floatation, and electro-deposition [21]. However, these methods may have drawbacks, such as incomplete metal removal, sludge generation, high reagent and energy requirements, and membrane fouling [22]. As a result, attention has turned towards biological remediation methods, such as biosorption, bioaccumulation, and biotransformation, which offer cost-effective and environmentally friendly solutions for efficiently removing heavy metals like chromium from industrial waste [23]. Microorganisms eliminate heavy metals through enzyme-catalysed metabolic pathways for toxic substance degradation, transforming them into carbon dioxide, methane, water, and biomass [24]. A wide range of microorganisms, bacteria [25], algae [26], fungi [27], and phyto species [28] have already been identified and isolated for potential heavy metal bioremediation and wastewater treatment. 

On the other hand, the current Directive 1999/31/EC of 26 April 1999 [29] on the landfill of waste, as well as Directive 2008/98/EC [30] on garbage [16], allows for the incineration or disposal of leather waste, chrome shavings, and solid waste in landfills. However, starting on 1 January 2025, the new Directive (EU) 2018/850 on landfills of waste [20] will prohibit landfilling or incineration as management methods for industrial textile waste. The European leather industry is increasingly focusing on sustainability initiatives to reduce the environmental impact and promote responsible sourcing [31] through the implementation of leather processing guides to improve and implement sustainable manufacturing practices [32], but most importantly, by developing more biodegradable leather products which could be composted. Leather biodegradability directly depends on the nature of the tanning agents and chemicals used in the manufacturing process; therefore, the leather industry has also focused on developing more biodegradable leathers with the aim of giving their end products a sustainable, non-contaminating second life or feasible disposal. These include the use of chrome-free leather manufacturing using non-metal tanning materials such as vegetable tannins (polyphenolic agents known to bind to collagen) [33], aldehyde compounds (phenolic synthetic compounds that bind to the amine groups of collagens, forming ionic bonds), calixarene [34], and other metal-free tanning agents [35]. Also, the development of faux leather substitutes, mainly consisting of polyvinyl chloride (PVC) or polyurethane (PU) layered onto a backing fabric (synthetic or natural, such as cotton or organic waste), undergo a surface coating procedure to boost their resilience and longevity. A recent study on the composting capacity of leathers [36] found that bovine leather treated with alginate derivatives degraded entirely within 21 to 25 days, conventionally produced wet blue leather degraded within 31 to 35 days, and vegetable-tanned bovine leather showed initial signs of degradation after 60 days but did not fully disintegrate even after 90 days. In contrast, alternative materials containing non-biodegradable components, like PU and PVC, showed no degradation after 90 days. 

The current compostability standard ISO 17088:2021 [37] was developed specifically for plastic compostability evaluation and has been commonly used to evaluate leather compostability. More specifically, ISO 14851:2019 [38] is used to determine the biodegradability potential of a plastic material by measuring oxygen consumption during 2–6 months in a closed bottle set up where the degradation percentage is calculated as the ratio of oxygen consumption to the theoretical oxygen demand (ThOD) [39]. Leather and plastic have different physico-chemical characteristics and show different degradation processes. Plastics degrade into smaller particles and pose a microplastic generation hazard, whereas leather poses chemical contamination hazards in the final compost. In this context, International Standard ISO 20136:2020, “Leather. Determination of degradability by micro-organisms” [40], was developed specifically for leather. This methodology uses a complex consortium of microorganisms from the tannery and urban wastewaters as the inoculum in liquid medium to measure the biodegradation potential of leather as a measure of the CO_2_ generated during leather degradation in 28 days. 

This article aims to present recent research discoveries regarding various front lines, the identification and isolation of microorganisms from tannery wastewaters, the identification of the microbial composition in the starting inoculum used for a leather biodegradation assay according to ISO 20136 (tannery and municipal wastewater), and the identification of microbial diversity shifts in the initial inoculum during the degredation process of different tanned leathers. The aim is to determine which genera are capable of biodegrading, with what type of tanning agent, and through which acts at the different degradation stages. 

The focus is on exploring their potential applications in enhancing leather biodegradability, wastewater treatment within the leather industry, and facilitating bioremediation processes for heavy metals. This study provides crucial insights into sustainable solutions for addressing environmental challenges within the leather industry, paving the way for the development of efficient and environmentally friendly treatments for tannery wastewater. It will also contribute to optimising leather composting processes, ultimately reducing the environmental impact of tannery operations and reinforcing the leather industry’s commitment to responsible and sustainable production.

## 2. Materials and Methods

### 2.1. Species Identification from Tannery Wastewaters

#### 2.1.1. Tannery Wastewater Collection and Preparation

A two-litre water sample was taken from an aeration pond at a wastewater treatment plant (Curtidos Serpiel S.A., Caudete, Spain) at a pH of 4.5 and an estimated chromium concentration of ~100 ppm. The water sample was filtered through a paper filter to remove macroscopic precipitates. Part of the filtrate was supplemented with 0.5% yeast to promote growth, and bacteriological agar was added up to 1%. A set of dilutions was prepared from the remaining filtered water up to 10^−4^ to the initial concentration. An amount of 100 μL of each dilution was used to seed the solid media initially prepared. Two plates were seeded for each dilution and incubated at room temperature for 10 days.

#### 2.1.2. Species Isolation

Colonies from the 10^−3^ dilution were resuspended in LB medium and reseeded on fresh plates at different dilutions. Up to 9 microorganisms were isolated and maintained through successive reseeding in standard LB medium. 

#### 2.1.3. Species Identification

Species identification was carried out by 16S/18S rRNA amplification and sequencing. Eight colonies were randomly selected and resuspended in 20 μL of ultrapure H_2_O, incubated at 99 °C for 15 min to lyse the cells. The NZY Microbial gDNA Isolation kit (NZYtech, Lisboa, Portugal) [41] was used to extract the DNA sample for the PCR with Taq DNA polymerase. Three pairs of degenerate universal oligonucleotides specific for the 16S/18S gene of bacteria, archaea, and eukaryotes were designed, including: BACF: 5′-AGAGTTTGATCCTGGCTCAG-3′—BACR: 5′-GGYTACCTTGTTACGACTT-3′—ARCF: 5′-TCCGGTTGATCCYGCBRG-3′—ARCR: 5′-TTMGGGGCATRCIKACCT-3′—EUKF: 5′-GGTTGATYCTGCCAGTAG-3′—EUKR: 5′-GTACACACCGCCCGTCGCT-3′. 

The PCR product obtained was sequenced using an automated sequencer and bacterial oligonucleotides, following two strategies: cloning the PCR products into the pGEM-T easy vector (Promega, Madison, WI, USA) and sequencing the PCR product directly after purifying the bands with the GFX PCR DNA and Gel Band Purification Kits (Cytiva, Marlborough, Massachusetts, USA) [42].

### 2.2. Microorganism Identification from Leather Biodegradation Assay

#### 2.2.1. ISO:20136:2020: Determination of Leather Degradability by Microorganisms’ Assay

A leather biodegradation assay (method B) was performed as described in ISO 20136:2020 [40]. The inoculum was a mixture of 50:50 (ratio) of tannery and municipal sewage wastewaters. Municipal wastewaters were collected from the local area treatment plant, and tannery wastewaters were collected from Curtidos Segorbe S.L. (Segorbe, Spain) [43], a different source of tannery wastewater than that mentioned in Section 2.1.1. Pure collagen from bovine Achilles tendon (Sigma-Aldrich^®^, St. Louis, Missouri, USA) [44] was used as a positive control (control sample). The assay was run for approximately 800 h (33 days). Five differently tanned leather samples were used for the assay; 0.5 g of each sample was placed in each Erlenmeyer flask containing minimal salts and the inoculum, as shown in Table 1. The total organic carbon content of the tested material was determined by elemental analysis. This allows the theoretical maximum quantity of carbon dioxide evolution to be calculated as biodegradation. 

#### 2.2.2. Wastewater and Leather Biodegradation Assay Sample Collection

Table 2 shows the samples taken from each Erlenmeyer flask, the hour at which they were withdrawn after the start of the assay, and the leather biodegradation stage registered at that time. According to the standard, either tannery or municipal wastewater can be used in the assay. Sample M1 was municipal residual wastewater, sample M2 was tannery wastewater, and sample M3 was a mixed inoculum (50:50) from M1 and M2. That was the inoculum used in the assay and the microorganism profile identified in samples M4 to M29 shift from this sample (M3). The aim of the microorganism identification in samples M1 and M2 was merely informative in confirming and comparing the final microorganism profile in sample M3. The remaining samples were drawn from an ongoing leather biodegradation assay (ISO 20136 assay) [45] at different stages of the degradation process (initial, exponential, and final) using a syringe and 0.1 microlitre filters. All samples were stored and transported at −20 °C.

#### 2.2.3. DNA Extraction and Quality Control

Cells were recovered from filters followed by DNA isolation using the Qiagen QIAsymphony PowerFecal Pro DNA Kit (Qiagen, Hilden, Germany) [46], which involved cellular lysis through mechanical disruption and enzymatic treatment. Subsequently, DNA purification from the sample was performed using a silica/gel column, allowing for DNA isolation and the removal of contaminants and inhibitors for future reactions using the QIAamp DNA Micro Kit (Qiagen, Hilden, Germany) [47]. Cellular lysis was carried out through mechanical disruption and enzymatic treatment. Subsequently, DNA purification from the sample was performed using a silica/gel column, enabling DNA isolation and the removal of contaminants and inhibitors for future reactions using the Qiagen QIAamp DNA Micro Kit. The quality and concentration of the DNA were evaluated using Nanodrop. 

#### 2.2.4. Sequence Library Preparation

An amount of 50 ng of the extracted DNA was amplified using a two-stage PCR protocol, the 16S Metagenomic Sequencing Library Preparation protocol Illumina 15044223. Primers were designed with the following structure: (1) an amplicon consisting of a universal linker sequence that allows for the incorporation of indices and sequencing primers using the Nextera XT Index kit [48]; (2) universal primers for the 16S rRNA gene [49]. 

#### 2.2.5. Sequencing

The sequencing libraries were prepared and loaded onto the Illumina MiSeq platform following a 300 bp × 2 paired-end design. In the second and final step of the assay, amplification indices were included. The resulting 16S libraries were quantified using fluorimetry with the Quant-iT™ PicoGreen™ dsDNA Assay kit (Thermo Fisher scientific, Waltham, MA, USA) [50]. Libraries were pooled before sequencing on the MiSeq platform (Illumina, Cambridge, UK) following 300 paired end-read design cycles. The size and concentration of the pool were evaluated using Agilent Bioanalyzer 2100 (Agilent, CA, USA) [51]. The PhiX Control library (v3) (Illumina) was combined with the amplicon library (with an expected percentage of 20%). The sequencing data became available after approximately 56 h. Image analysis, base calling, and data quality control were performed on the MiSeq platform (MiSeq Control Software (MCS v3.1)). The raw sequences, forward (R1) and reverse (R2), were merged to obtain the complete sequence using the BBMerge package of the BBMap software V.38. With this approach, the ends of the sequences overlapped to obtain complete sequences. Sequencing adapters were searched for and removed using the Cutadapt program (v 1.8.1) to reduce bias in the subsequent annotation stage.

#### 2.2.6. Bioinformatic Analysis

The identification and removal of non-genomic regions or those with poor quality was carried out. Initially, the BBMerge module of BBMap software V.38 was employed to merge each pair of sequences (R1 and R2) from the sequencing platform, ensuring a minimum overlap of 70 nts at each end. This process resulted in a unique and complete sequence. Subsequently, the Cutadapt v1.8.1 program was used to detect and eliminate sequencing adapters from both ends in each sample. Once the adapter-free sequences were obtained, reads with a quality below Q20 and lengths of less than 200 bp were removed. The Reformat module of BBMap V.38 was utilised for this analysis, enabling the trimming of nucleotides with a quality value below Q20 from both ends.

The final step in the quality processing involved eliminating potential chimaera sequences resulting from incomplete extension during the PCR amplification. This step was performed using the cd-hit-dup module, part of the cd-hit 4.8.1 software, predicting these amplicons de novo from the sample. The resulting sequences were then used for annotation. Sequences sharing 99% similarity were grouped into a single sequence using the “cd-hit” program. The outcomes were applied to the sequence group represented by the analysed one, referred to as the Operational Taxonomic Unit (OTU). Subsequently, each sequence group was compared against the RefSeq 16S rRNA gene database (NCBI) using the local alignment BLAST strategy to associate each group with a taxonomic group from the database. 

## 3. Results

### 3.1. Species Identification from Tannery Wastewater Treatment Plant (Curtidos Serpiel S.A., Caudete, Spain)

#### Species Identification

Agarose gel electrophoresis (1%) of the PCR products, as shown in Figure 1, was obtained from eight colonies using oligonucleotides for bacteria, archaea, and eukaryotes. The bands seen from B1 to B7, marked with a red box in Figure 1 are the same size as those seen in Figure 2. 

All PCR products were sequenced with the corresponding oligonucleotides at the Genomics and Proteomics Service of the University of Alicante Technical Services. The sequences obtained in each case were compared using the NCBI databases and EZBioCloud. The identified bacterial strains are shown in Table 3. Strains were identified by sequencing the whole bacterial 16S rRNA gene; therefore, species indicating a similarity percentage of over 99% were considered to be an exact match [53]. Other species would have to be identified by whole-genome sequencing. 

### 3.2. Microorganism Identification from Leather Biodegradation Assay Using ISO 20136:2020 

#### 3.2.1. ISO:20136: Leather Determination of Degradability by Microorganisms

Biodegradation results for the leather samples described in Section 2.2.1 are shown in Figure 3. These biodegradation curves show the percentage of leather biodegradation throughout the assay (30 days); the times shown in Table 2 (Section 2.2.2) correspond to the time (hours) in this graph. There is an initial, exponential, and final phase of leather biodegradation for all the samples at different levels of the biodegradation percentage. Collagen is used as a positive control, since it fully degrades in approximately 30 days [40]. At the final stage of the assay (747 h since time 0), when the last samples were taken, the biodegradation percentage for collagen was 81.5%, oxazolidine was 59.4%, glutaraldehyde was 5.2%, chromium was 7.6%, and aluminium was 23.2%. 

#### 3.2.2. Sequencing

The concentration and purity of the extracted DNA reached enough levels to proceed with the preparation of sequencing libraries. The sequencing results, including the number of raw sequences obtained, average length, total sequenced bases measured in Megabases (Mb), and the mean quality for forward (R1) and reverse (R2) sequences. The read count exceeded 50,000 in all analysed samples, except for sample 210527A-M4, which yielded 43,447 reads. 

#### 3.2.3. Bioinformatics and Species Identification

Although this sample did not reach the desired read count during the sequencing process, this quantity proved optimal, as it reached the plateau zone in the rarefaction curve. The profiles were studied at a maximum genus level, providing species-level information for discussion on potentially implicated species. However, due to the product size of around 400 nts for bacteria, precise species identification cannot be guaranteed.

It can be determined that it is close to the plateau zone with the inspected number of sequences and the microbial profile. As shown in the rarefaction curves (alpha-diversity index) in Figure 4, most samples are in the plateau zone, indicating that an increase in sequences would not yield a significant rise in the number of newly detected genera.

The statistical analysis employing the vegan package in R [54] allows for the examination of the organism–abundance relationship. These results are presented in Table 4. The Shannon index estimates the specific biodiversity present in the sample, providing a positive numeric value starting from zero (indicating a single species) and increasing as diversity increases [55]. The Chao 1 value estimates the total number of species that may be present in the sample based on the number of less-represented species in the sample [56]. In taxon-based approaches, accurately estimating the number of microbial species in a sample is challenging due to the complexity of microbial diversity; therefore, assessing species richness is crucial for understanding biological communities effectively [57]. Richness estimators like Chao 1 are employed to deduce the total richness of a microbial community based on observed Operational Taxonomic Units (OTUs). Unlike rarefaction, which compares observed richness among samples, richness estimators predict total richness from a single sample. For all the samples, the Shannon index is around four and the Chao 1 index was around 2500. Sample M4 is the only one that differs from the other samples, showing a Shannon index of 2.70 and a Chao 1 of 266. 

Local BLAST alignment is conducted to associate each of the obtained reads (after cleaning and filtering) with an organism. Working with a well-curated database is crucial, as databases may be incomplete or lack certain organisms due to being sequenced or not being taxonomically assigned. In cases where a sequence is associated with multiple hits with the same e-value and identity, the first hit is considered the best. It is important to note that these taxonomic identifications represent the best outcome in local alignment, not guaranteeing the organism’s presence in the sample.

Short sequences are excluded from the analysis to reduce the likelihood of duplicating sequences in the database and encountering false positives or misassignments. Sequencing errors that can alter assignments to a specific organism by modifying sequence similarity are also eliminated from the analysis. For each sequence, details such as the percentage identity with the database hit, e-value, absolute number of identical positions in the alignment, relative number of identities considering the total number of alignment bases, absolute number of gaps in the alignment, and relative number of gaps considering the total number of alignment bases are provided. 

Once each read is associated with an organism, various analyses and filters are applied. Reads are grouped based on family, genera, and species, with associations typically made at the family or genera level, but species-level associations are also provided. The database is curated according to different taxonomic levels described in the NCBI database [58]. Some levels may remain null if they are not characterised, cultivated, or unknown. The presence of different taxonomic groups in the analysed samples is summarised in tables and graphs. A rarefaction curve is generated for each analysed sample, comparing the number of analysed sequences with the number of detected taxa at different levels, as shown in Figure 4. This curve helps determine if the detection has reached saturation, indicating that all organisms have been detected or if more sequences are needed to capture the total variability of the sample. Sample M4 (collagen at 17% biodegradation at 52 h since the start of the assay) has shown to be the only one that differs from other samples, showing a Shannon index of 2.70 and a Chao 1 of 266, considerably lower than the rest of the reads. This is also shown in Figure 4, where the alpha-diversity index of M4 is considerably lower than the rest of the samples. The reduction in these is caused by the immense presence of *Acinetobacter* in the sample. This bacterium shows to be a great collagen degrader, as its presence grows exponentially within the first 48 h within the assay. On the other hand, M3, being the mix of both wastewaters, shows the highest alpha-diversity index as well as the highest Shannon index and Chao parameters, at five and five-hundredths and 3383, respectively. 

The bacterial abundance detected in the analysed samples at the genera level is shown in Figure 5; the M1–M29 samples refer to those shown previously in Table 2 and Table 4. Additionally, this diagram incorporates the LCBD beta diversity index, revealing diversity patterns. High LCBD values indicate that the bacterial composition of the sample significantly differs from the rest of the studied samples [59]. Bacterial strains identified at the highest percentages in municipal wastewaters (sample M1) were *Nakamurella endophytica* (10.76%) [60], *Clostridium saudiense* (9.87%) [61], *Romboutsia timonensis* (9.21%) [62], and *Mycolicibacterium peregrinum* (3.41%) [63]. On the other hand, the bacterial strains most present in tannery wastewaters (sample M2) were *Pseudonocardia rhizophila* (13.02%) [64], *Variibacter gotjawalensis* (11.46%) [65], *Pseudorhodoplanes sinuspersici* (6.42%) [66], *Hyphomicrobium aestuarii* (3.46%) [67], and *Mycolibacterium peregrinum* (3.38%) [63]. The inoculum used for the assay was sample M3, a 50:50 mixture of M1 and M2, and, as would be expected, the composition in terms of bacterial presence is halfway between M1 and M2. 

For comparison purposes, the results of bacterial identification have been categorised according to the different tanned leather samples. This allows for a comparison between different tanning agents and various stages of biodegradation within the same type of leather. M1, M2, and M3 have been left in all the following bar plot graphs, since they represent the initial bacterial composition of the inoculum. M3 was the inoculum used in the assay; therefore, all the bacterial shifts in the following samples diverge from sample M3. Figure 6a shows the bacterial abundance at the species level detected in the analysed samples for collagen (control sample). As collagen biodegradation takes place (samples M4–M25), the most abundant species present are *Brevundimonas terrae*, *A. johnsonii*, and *M. peregrinum*. Figure 6b shows a line graph representing bacterial presence shift in the sample as a percentage of the top six bacterial species identified at the different stages of collagen biodegradation. Within the first 52 h after the start of the assay, the most predominant genus is *Acinetobacter.* Some species, such as *Chryseobacterium indoltheticum*, start at very high concentrations but quickly decrease, as they cannot proliferate using collagen as the only carbon source. 

Figure 7a shows the bacterial abundance at the species level detected in the analysed samples for chromium-tanned leather. As leather biodegradation occurs (samples M4–M25), the most abundant species are *Brevundimonas terrae*, *Acinetobacter johnsonii*, and *Mycolicibacterium peregrinum*. Figure 7b shows a line graph representing bacterial presence shift in the sample as a percentage of the top six bacterial species identified at the different stages of chromium leather biodegradation. In this case, *Brevundimonas* is the predominant bacterial genus; *Brevidumonas terrae* is shown to be the bacterial strain most present in samples M11 to M4, shifting to *Brevundimonas kwangchunensis*, which is present at 15.9% in the sample.

Figure 8a shows the bacterial abundance at the species level detected in the analysed samples for glutaraldehyde-tanned leather. As leather biodegradation takes place (samples M10–M27), the most abundant species present are *A. johnsonii*, *B. terrae*, and *M. peregrinum*. Figure 8b shows a line graph representing bacterial presence shift in the sample as a percentage of the top six bacterial species identified at the different stages of glutaraldehyde leather biodegradation. In this case, *A. johnsonii* is the predominant bacterial strain, being present in more than 10% of all the samples from M10 to M27.

Figure 9a shows the bacterial abundance at the species level detected in the analysed samples for oxazolidine-tanned leather. As leather biodegradation occurs (samples M6–M26), the most abundant species present are *B. terrae*, *A. johnsonii*, and *M. peregrinum*. Figure 9b shows a line graph representing the bacterial presence shift in the sample as a percentage of the top six bacterial species identified at the different stages of oxazolidine leather biodegradation. In this case, *B. terrae* is the predominant bacterial strain in all samples from M6 to M26, always present at around 20% in all the samples. 

Figure 10a shows the bacterial abundance at the species level detected in the analysed samples for aluminium-tanned leather. As leather biodegradation takes place (samples M7–M29), the most abundant species present are *B. terrae*, *A. johnsonii*, and *M. peregrinum*. Figure 10b shows a line graph representing the bacterial presence shift in the sample as a percentage of the top six bacterial species identified at the different stages of aluminium leather biodegradation. In this case, *B. terrae* is the predominant bacterial strain in all samples from M6 to M26, always present at around 20% in all the samples. 

## 4. Discussion

### 4.1. Species Identification from Tannery Wastewaters 

Over the years, there have been different standards for fixation for the use of 16S rRNA genes in taxonomy identification [68]. The latest cutoff value at the species level has been evaluated at 98.7% [69]; however, several authors have shown that these thresholds do not apply to multiple genera [70]. Isolated species 1, *Dietzia maris*, has been previously isolated from the soil for zinc bioremediation [71], for petroleum hydrocarbons and crude oil degradation [72,73], and for tolerance to heavy metals such as cadmium or cobalt [74] and multiple extreme resistance [75]. Species 2, *T. pasteurii*, has been previously identified for the reduction in hexavalent chromium [76], showing the following two potential pathways for Cr (VI) removal: the sulphidogenesis-induced Cr (VI) reduction pathway, with 90% Cr (VI) removal by sulphide generated from the biological reduction in sulphate, and direct 10% Cr (VI) removal by bacteria as the electron acceptor [77]. *Corynebacterium lubricantis* has been previously isolated from the chromite mine seepage of Odisha as a heavy metal-tolerant and chromate-reducing bacterium [78], as well as isolated from a chromium-polluted soil, and tested for its chromate-reduction capability and multiple heavy metal tolerance up to a concentration of 22 mM [79]. *Microbacterium* strains have been previously studied for chromium waste biocementation [80] and isolated from tannery wastewater for hexavalent chromium reduction [81]. *B. safensis* has been isolated from tannery effluent as a chromium (Cr) and tannic acid (TA) resistance bacterial strain [82], isolated from contaminated coal mining soil for chromium reduction [83], and isolated from rare-earth ore for hexavalent chromium conversion to trivalent chromium, where the gene *nfrA* is involved [84]. All of these species represent good candidates for further investigation, in which they are submitted to stress conditions and evaluated for functional groups in heavy metal bioremediation. 

The identification of the specific functional groups responsible for metal ion binding to microbial biomass is crucial for understanding the biosorption mechanism in efficient methods for removing heavy metals from contaminated environments [85]. The type, structure, and arrangement of functional groups can vary significantly between microorganisms [86], with many of these groups primarily identified on microbial cell walls [87]. Functional groups such as aldehydes, alkyl chains, amides, amines, alcohols/phenols, carboxylic acids, esters, organic halides, phosphates, sulphoxides, and aliphatic organic chains of cellulose have been identified as key players in the biosorption of chromium [88]. Multiple spectroscopic and microscopic techniques, such as infrared and Raman spectroscopy, electron dispersive spectroscopy, and nuclear magnetic resonance (NMR), have been identified for active sites involved in binding heavy metal ion identification [89]. However, most studies employ the Fourier transform infrared spectroscopic (FT-IR) technique to identify and characterise certain functional groups present in microbial biomass to uptake toxic heavy metals such as hexavalent chromium [90]. Specific functional groups and mechanisms for Cr biosorption have already been identified for certain species, such as *Bacillus marisflavis*, *Bacillus arthrobacter* [91], and *Klebsiella* sp. [91], which include OH, the –NH acetamido group, amide bonds, the C=O of COO-, free phosphates, phosphate groups, –CN and NH_2_, O-H, -CONH-, -COOH, C=C, and -CH_2_ (Freundlich adsorption isotherm), respectively.

### 4.2. Microorganism Identification from Leather Biodegradation Assay

#### 4.2.1. ISO:20136:2020: Leather Determination of Degradability by Microorganisms

The leather biodegradation process heavily depends on the tanning agent employed, which gives leather specific physical–chemical properties [1]. Pure collagen’s exponential degradation phase was shown to be in the first 16 h of the assay. Oxazolidines are cyclic condensation products of β-amino alcohols, aldehydes, or ketones [65]. The ease of the hydrolysis of oxazolidines in aqueous solution relates directly to their structural features, a cyclic ring structure that contains both a nitrogen atom and a carbonyl group [66]. These functional groups are susceptible to nucleophilic attack during hydrolysis, leading to the cleavage of the ring and the formation of the corresponding β-amino alcohol and aldehydes or ketones [67]. This leads to a higher biodegradation potential, like mineral-based tanning methods such as zeolites, which have shown an 81% biodegradation potential in 30 days [68]. Metal-tanned hides form stable crosslinks between the collagen fibres, rendering the material less susceptible to enzymatic attack by microorganisms. These can exhibit toxicity to microorganisms, inhibiting their ability to degrade the leather [69]. The presence of toxic metal ions can disrupt microbial activity, slowing down the biodegradation process [70]. Recent studies have shown that chromium-free tanned leather has a higher degree of biodegradation and a faster biodegradation rate than chrome-tanned leather. These studies have established that increasing the tanning effects through enhancing tanning methods or conditions decrease the leather biodegradability [39]. 

#### 4.2.2. Bioinformatics and Species Identification

More than 10,000 bacterial species have been identified to be present within all the analysed samples of the leather biodegradation assay, and 7 bacterial strains have been isolated from tannery wastewaters. Bacterial genera identified and isolated from the first tannery wastewater sample, as described in Table 3, have been identified in low percentages in sample M2 (tannery wastewater), as follows: *D. maris* (0%), *T. pasteurii* (0.012%), *C. lubricantis* (0%), *M. laevaniformans* (0%), and *B. safensis* (0%). Tannery wastewaters are generally characterised by a dark brown colour and high levels of pollutants, total dissolved solids (TDSs), chromium, phenolics, and a high pH [92]. Bacterial diversity differences amongst these samples could be due to the changes in these conditions, which vary depending on which stage and time the sample was taken and the type of tanning that was performed [93]. 

The most identified microorganisms throughout the assay are *Acinetobacter*, *Brevundimonas*, and *Mycolicibacterium*. The *Acinetobacter* genus is frequently found in aquatic environments such as wastewaters and river waters [94]. Four *Acinetobacter* strains (*A. johnsonii, A. modestus, A. tjernbergiae,* and *A. junii*) have been identified. These species have previously been identified and isolated from tannery and residual wastewaters, characterised and evaluated for chromium bioremediation [95,96,97,98] and Cr^6+^ transformation to Cr^3+^ [99,100]. *Brevundimonas* have been mainly identified and isolated from sedimented waters [101], rivers [102], and soil samples [103,104,105]. *Brevundimonas* strains have also been identified and isolated for heavy metal bioremediation [106,107], cadmium and zinc bioremediation [108], and arsenic resistance [109,110]. *Mycolicibacterium* is a non-tuberculous identified and isolated in cotton fields [111], peat bogs [112], sea coasts [113], and mangrove sediments [114]. This strain has been previously identified and isolated for zinc–lead bioremediation [115]. 

Within the first 16 h since the start of the assay, at the exponential phase of collagen biodegradation (M4 and M5), 60% and 37% of the inoculum consists of the genus *Acinetobacter* along with *Brevundimonas*, present at 6.75% and 16%, respectively. Within the bacterial profile for chromium samples, *Variibacter gotjawalensis* is present at over or around 5% in all samples and at 11.6% in sample M28 at the final stages of the biodegradation assay (7.5% of leather biodegradation). This strain only appears significantly in samples from the chromium-tanned leather assay. This strain has been previously isolated from soil in a lava forest in Korea [65] and tested in Cd-Zn-Pb-contaminated soil for phytoextraction [116]. 

It must be noted that these identified genera (*Acinetobacter*, *Brevundimonas*, and *Mycolicibacterium)* are merely present in the initial inoculum (M3), where *Mycolicibacterium* is at around 5%, and the other two do not reach 1%. There is an immense jump to higher percentages as the leather biodegradation occurs. The amount of leather sample and, therefore, tanning agents are not significant enough to evaluate the capability of the identified species to degrade and/or tolerate these molecules in higher concentrations from these results. It is clear from the results that they have the capability of bond breakage between the tanning agent and the collagen fibres, and biodegrade collagen itself. For certain tanning agents such as glutaraldehyde (Figure 8b) and oxazolidine (Figure 9b), *Acinetobacter johnsonii* and *Brevundimonas terrae,* respectively, are the present species, and therefore the most active species in terms of leather biodegradation. Further investigations would have to be carried out to gain a deeper understanding of the species’ biodegradation mechanisms.

## 5. Conclusions

This study has contributed to understanding the microbial diversity and abundance in the context of leather biodegradation and heavy metal resistance. Despite the great abundance of the identified bacterial species, the identification of bacterial genera such as *Acinetobacter*, *Brevundimonas*, and *Mycolicibacterium* in the samples has provided valuable insights into the potential microbial candidates, showcasing their potential applications in enhancing leather biodegradability, wastewater treatment, and bioremediation processes for heavy metals.

Overall, the findings of this study underscore the importance of sustainable solutions for addressing environmental challenges within the leather industry. The selective isolation and identification of microorganisms with dual capabilities offer promising prospects for developing efficient and environmentally friendly treatments for tannery wastewater, leather biodegradation, and heavy metal remediation, thereby contributing to the advancement of sustainable practices in the leather industry.

## Figures and Tables

**Figure 1 microorganisms-12-01029-f001:**
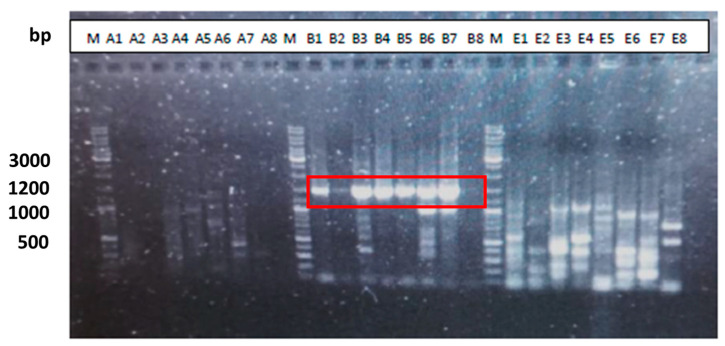
M: GeneRuler DNA Ladder Mix (Thermo Fisher scientific) [52]. A1–A8: PCR products using archaea oligonucleotides. B1–B8: PCR products using bacteria oligonucleotides. E1–E8: PCR products using eukaryotes oligonucleotides.

**Figure 2 microorganisms-12-01029-f002:**
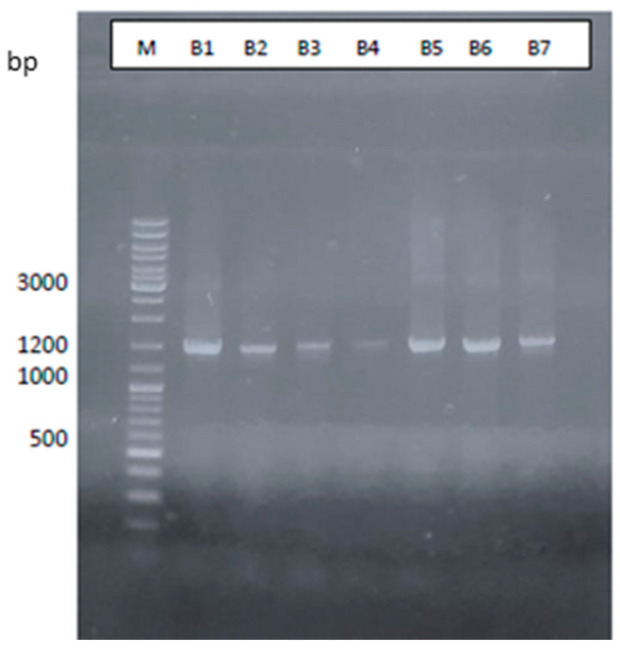
Agarose gel electrophoresis at 1% of the PCR products purified from PCR reactions performed using oligonucleotides for bacteria. B1–B7: PCR products using bacteria oligonucleotides for 16S rRNA. Band sizes in bp are indicated. M: GeneRuler DNA Ladder Mix (Thermo Fisher scientific) [49].

**Figure 3 microorganisms-12-01029-f003:**
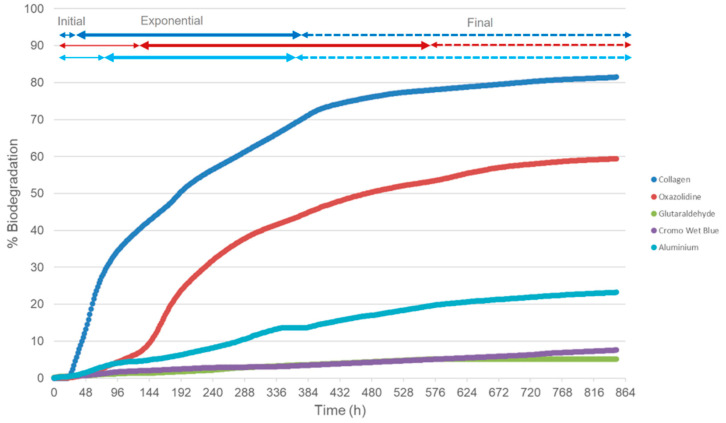
Biodegradation results of the ISO 20136 assay performed with four different types of leather samples. The graph shows the biodegradation percentage vs. time of each sample. Collagen is the positive control. At the top of the graph, initial (thin line), exponential (thick line), and final (dotted line) biodegradation phases are shown for collagen, oxazolidine, and Aluminium in matching colour to the graph.

**Figure 4 microorganisms-12-01029-f004:**
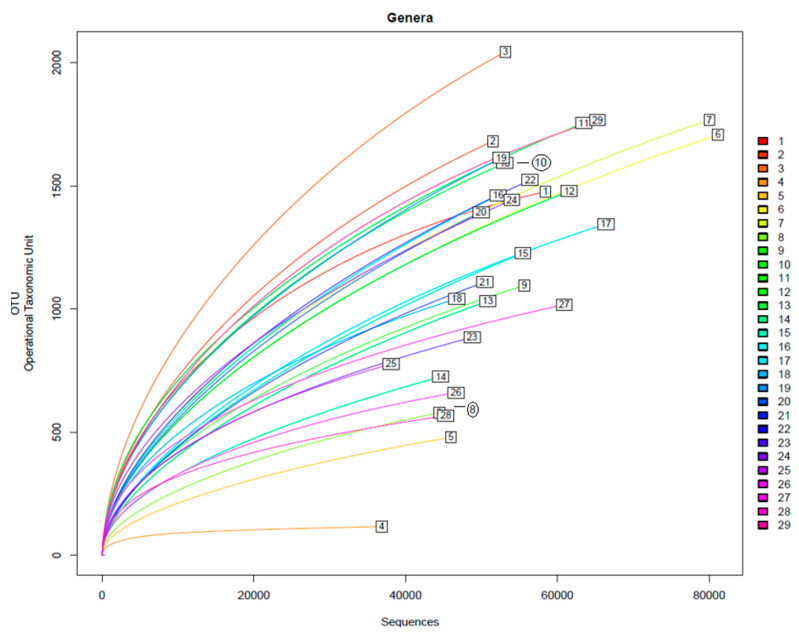
Rarefaction curves of the amplified samples for bacteria detection. Samples shown are from M1 to M29.

**Figure 5 microorganisms-12-01029-f005:**
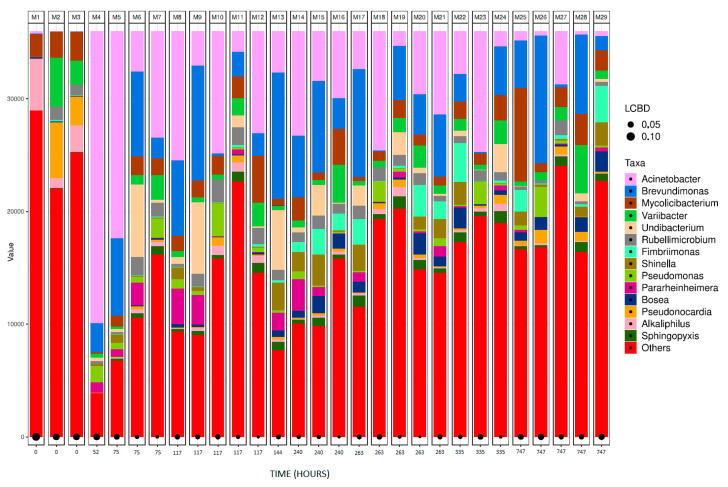
Bar plot figures representing the proportions of detected bacterial genera in the studied samples.

**Figure 6 microorganisms-12-01029-f006:**
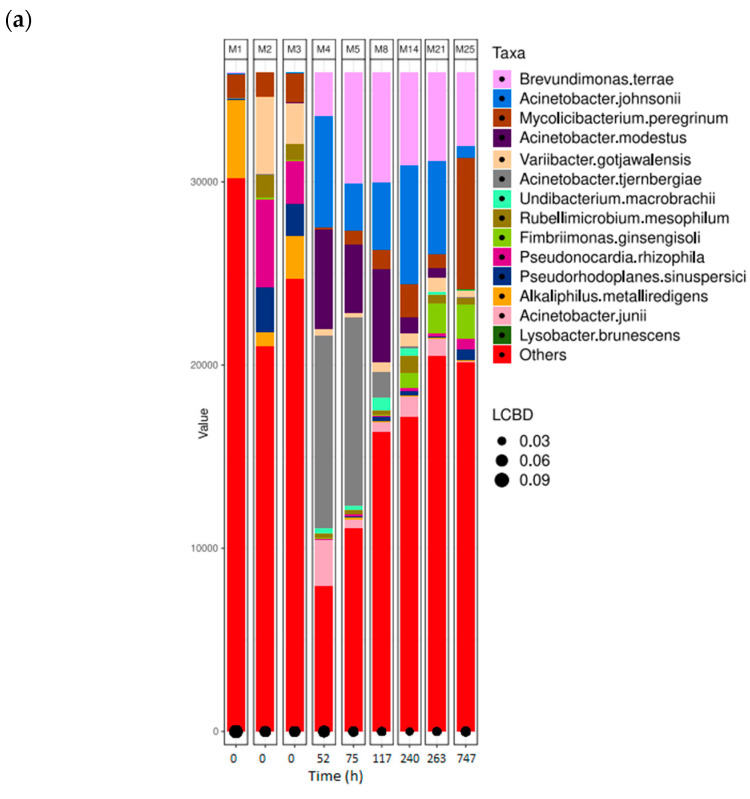
Bacterial species detected for collagen samples in ISO 20136. (**a**) Bar plot figures representing the proportions of detected bacterial species in the studied samples (M4, M5, M8, M14, M21, and M25). Sample M1 was the municipal residual wastewater, sample M2 was the tannery wastewater, and sample M3 was a mixed inoculum (50:50). (**b**) Line graph representing the bacterial shift as the bacterial presence in the sample (%) of the top six bacterial species found for all collagen samples.

**Figure 7 microorganisms-12-01029-f007:**
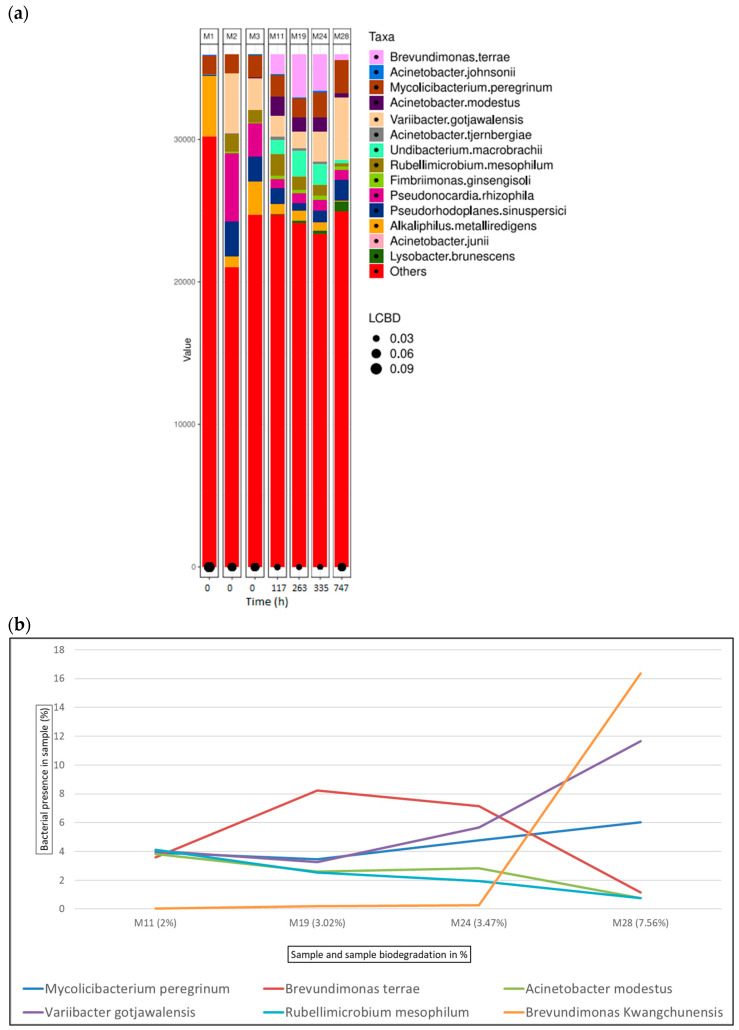
Bacterial species detected for chromium samples in ISO 20136. (**a**) Bar plot figures representing the proportions of detected bacterial species in the studied samples (M11, M19, M24, and M28). Sample M1 was the municipal residual wastewater, sample M2 was the tannery wastewater, and sample M3 was a mixed inoculum (50:50). (**b**) Line graph representing the bacterial shift as the bacterial presence in the sample (%) of the top 6 bacterial species found for all chromium samples.

**Figure 8 microorganisms-12-01029-f008:**
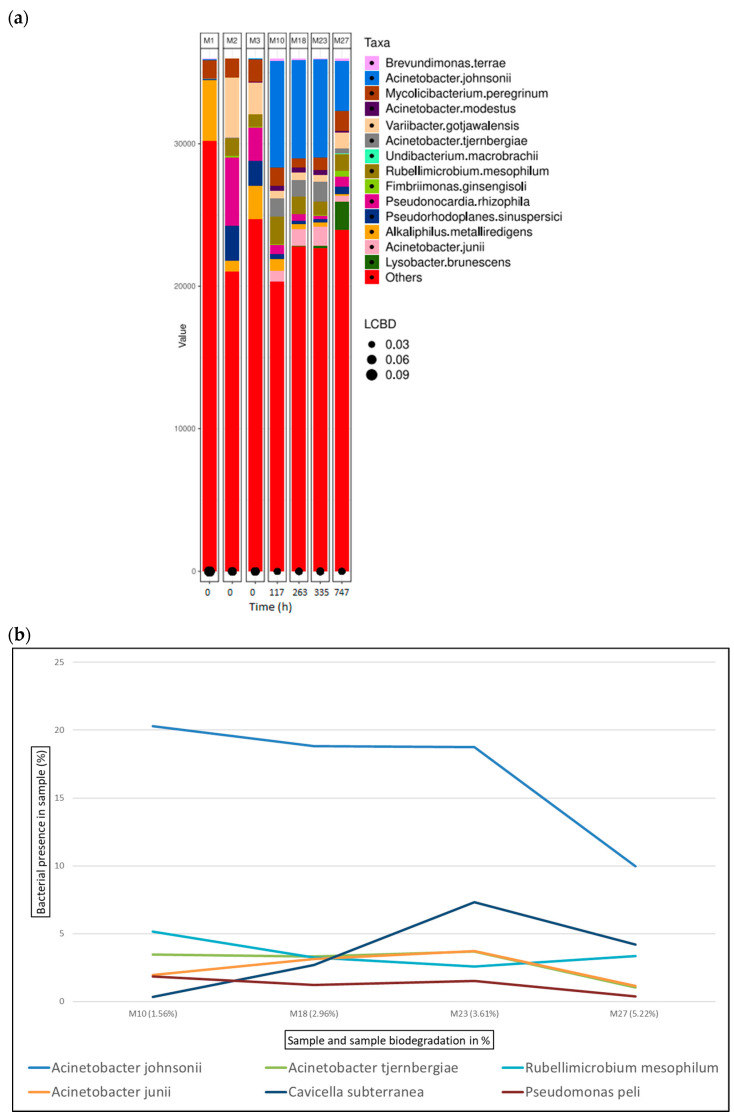
Bacterial species detected for glutaraldehyde samples in ISO 20136. (**a**) Bar plot figures representing the proportions of detected bacterial species in the studied samples (M10, M18, M23, and M27). Sample M1 was the municipal residual wastewater, sample M2 was the tannery wastewater, and sample M3 was a mixed inoculum (50:50). (**b**) Line graph representing the bacterial shift as the bacterial presence in the sample (%) of the top six bacterial species found for all glutaraldehyde samples.

**Figure 9 microorganisms-12-01029-f009:**
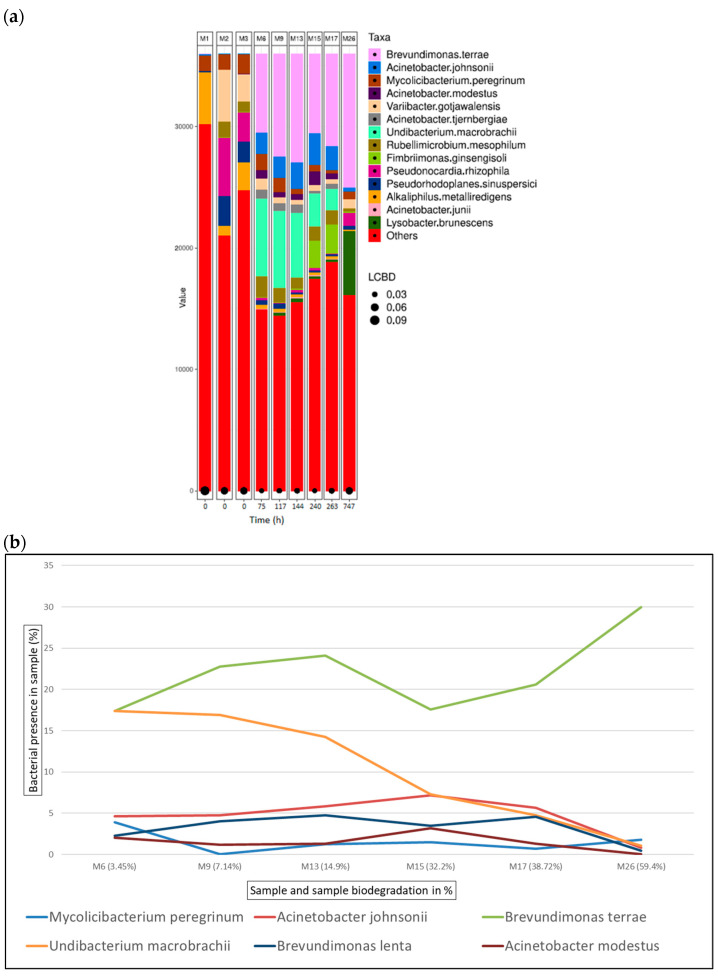
Bacterial species detected for oxazolidine samples in ISO 20136. (**a**) Bar plot figures representing the proportions of detected bacterial species in the studied samples (M6, M9, M13, M15, M17, and M26). Sample M1 was the municipal residual wastewater, sample M2 was the tannery wastewater, and sample M3 was a mixed inoculum (50:50). (**b**) Line graph representing the bacterial shift as the bacterial presence in the sample (%) of the top six bacterial species found for all oxazolidine samples.

**Figure 10 microorganisms-12-01029-f010:**
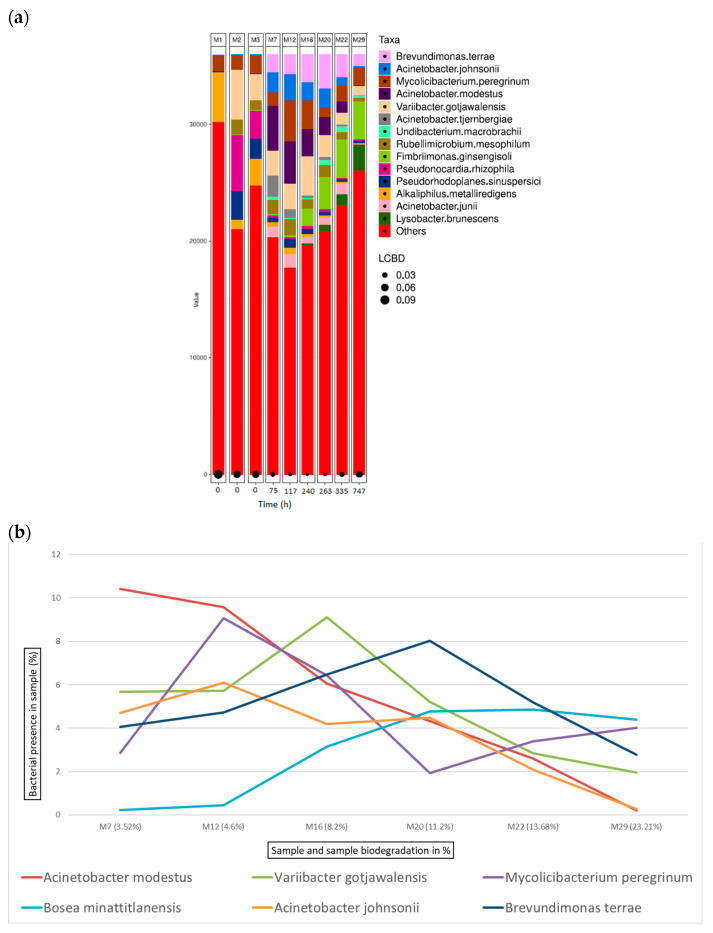
Bacterial species detected for aluminium samples in ISO 20136. (**a**) Bar plot figures representing the proportions of detected bacterial species in the studied samples (M7, M12, M16, M20, M22, and M29). Sample M1 was the municipal residual wastewater, sample M2 was the tannery wastewater, and sample M3 was a mixed inoculum (50:50). (**b**) Line graph representing the bacterial shift as the bacterial presence in the sample (%) of the top six bacterial species found for all aluminium samples.

**Table 1 microorganisms-12-01029-t001:** Leather samples (S1 to S4) and the control (pure collagen) used in the leather biodegradation assay.

Sample	Tanning Agent	Carbon%	Weight (g)	Erlenmeyer Flask Ref.
Control	None	50.60	0.5047	2
S1	Oxazolidine	44.76	0.5006	4
S2	Glutaraldehyde	47.76	0.5002	7
S3	Chromium	36.11	0.5012	10
S4	Aluminium	41.45	0.5036	14

**Table 2 microorganisms-12-01029-t002:** Extracted samples from an ongoing leather biodegradation assay.

Sample	Time (h) ^1^	E. Flask Ref.	Leather Sample	Biodegradation (%)
M1	0	-	None	0
M2	0	-	None	0
M3	0	-	None	0
M4	52	2	Control	17
M5	75	2	Control	32
M6	75	4	S1	3.45
M7	75	14	S4	3.52
M8	117	2	Control	40.5
M9	117	4	S1	7.14
M10	117	7	S2	1.56
M11	117	10	S3	2
M12	117	14	S4	4.6
M13	144	4	S1	14.9
M14	240	2	Control	57
M15	240	4	S1	32.2
M16	240	14	S4	8.2
M17	263	4	S1	38.72
M18	263	7	S2	2.96
M19	263	10	S3	3.02
M20	263	14	S4	11.2
M21	263	7	S2	62.2
M22	335	14	S4	13.68
M23	335	7	S2	3.61
M24	335	10	S3	3.47
M25	747	2	Control	81.5
M26	747	4	S1	59.4
M27	747	14	S4	5.22
M28	747	10	S3	7.56
M29	747	14	S4	23.21

^1^ Time (h) since time 0 of the assay Biodegradation stage at which the sample was extracted. M1: municipal wastewater, M2: tannery wastewater, M3: 50:50 mixture of M1 and M2.

**Table 3 microorganisms-12-01029-t003:** Bacterial strains identified from tannery wastewater.

Name	Top-Hit Taxon	Similarity (%)	Completeness (%)	Length (bp)
Species 1	*Dietzia maris*	99.48	94.4	1355
Species 2	*Trichococcus pasteurii*	99.21	94.3	1396
Species 3	*Corynebacterium lubricantis*	97.86	97.7	1034
Species 4	*Microbacterium laevaniformans*	99.47	95.8	1370
Species 5	*Bacillus safensis*	99.36	96.2	1416
Species 6	*Proteiniphilum AB243818_s*	99.26	98	1419
Species 7	*Proteiniphilum AB243818_s*	95.80	97	1405

**Table 4 microorganisms-12-01029-t004:** Bacterial diversity for each sample according to the Shannon and Chao 1 parameters.

Sample	Shannon	Chao 1
M1	4.56	2127
M2	4.62	2816
M3	5.05	3383
M4	2.70	266
M5	3.07	1205
M6	4.05	3296
M7	4.65	3177
M8	3.68	1344
M9	3.79	2232
M10	4.59	2817
M11	5.24	3295
M12	4.55	3053
M13	3.77	2284
M14	3.96	1821
M15	4.16	2818
M16	4.68	3068
M17	4.10	2553
M18	4.33	2040
M19	5.11	3406
M20	4.72	2972
M21	4.16	2198
M22	4.84	3049
M23	4.10	1837
M24	4.99	3031
M25	4.01	1506
M26	3.32	1217
M27	4.73	1903
M28	4.15	1105
M29	4.94	2727

## Data Availability

The raw data supporting the conclusions of this article will be made available by the authors on request.

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
