# Peer review of "Selective Isolation and Identification of Microorganisms with Dual Capabilities: Leather Biodegradation and Heavy Metal Resistance for Industrial Applications"

_microorganisms, 2024, doi:10.3390/microorganisms12051029_

Round 1

Reviewer 1 Report

Comments and Suggestions for Authors

The manuscript describes the microbial diversity changes during the biodegradation process of different treated-leather samples. The metagenomic approach led to the identification of the most representative bacterial species found in each experimental condition and the change in representative during the leather biodegradation process. The manuscript includes a large amount of experimental data with some relevance. The manuscript could interest Microorganisms journal readers, but many aspects must be improved.

The main concerns in the manuscript are listed below:

The manuscript's title does not describe the experimental set or the study's results. Some bacterial species were isolated, but no evidence of the relevance of such bacterial isolates for the leather biodegradation process was determined in the study. Moreover, how heavy metal resistance was established, but no experiments for heavy metal resistance were carried out in the study.

The description of the leather degradation experiments is difficult to understand; treatments evaluated are presented mixed, and why different sampling times and volumes were chosen? It would have been better to organize the data by type of treatment.

If nine colonies were isolated, why just eight were analyzed for identification?

In figure 1, indicate which DNA bands are expected or those of interest

In Figure 3, it is not clear which microorganisms were used in the experiment

The conclusions seens superficial.

Conclusions do not mention the potential of selectively isolated bacteria, it is not clear how the identified microorganisms can have a dual capacity, leather biodegradation and resistance to heavy metals. How the evaluation of the microbial communities present during the degradation process can improve leather biodegradation processes, what is the applicability of the data generated.

The manuscript has several technical/format issues to correct.

Author Response

Reviewer 1. Comments and Suggestions for Authors

The manuscript describes the microbial diversity changes during the biodegradation process of different treated-leather samples. The metagenomic approach led to the identification of the most representative bacterial species found in each experimental condition and the change in representative during the leather biodegradation process. The manuscript includes a large amount of experimental data with some relevance. The manuscript could interest Microorganisms journal readers, but many aspects must be improved.

The main concerns in the manuscript are listed below:

The manuscript's title does not describe the experimental set or the study's results. Some bacterial species were isolated, but no evidence of the relevance of such bacterial isolates for the leather biodegradation process was determined in the study. Moreover, how heavy metal resistance was established, but no experiments for heavy metal resistance were carried out in the study:

Response: The ultimate goal of our investigation is to identify microorganisms capable of degrading leather and resisting heavy metals. We began by isolating and characterizing bacterial strains from tannery wastewaters, known for their high levels of chromium, aluminum, and other metals and significant hide residues from tanning. Our ongoing research involves assessing these strains' ability to withstand heavy metals, degrade leather, and potentially contribute to composting processes. While detailed results are not presented here, our isolation conditions—where tanning tank water consists entirely of hides and high metal concentrations—suggest these isolates possess both leather-degrading and heavy metal-resistant properties.

The description of the leather degradation experiments is difficult to understand; treatments evaluated are presented mixed;

why different sampling times:

Response: As biodegradation occurs for each leather type, samples are taken from the assay. Each type of leather and pure collagen (used as the control) have different biodegrading times. Therefore, samples are taken according to the biodegradation percentage shown in the assay. Samples were taken, so we had samples for all leather samples' initial, exponential, and final stages of biodegradation. The work investigates how microbial diversity changes as the different leather samples are degraded. Therefore, M3 is the initial microbial diversity found in the assay, and the rest of the samples (arranged according to tanning agent) represent microbial diversity shifts as tanned leather is biodegraded.

 and volumes were chosen? It would have been better to organize the data by type of treatment.

Response: The volume column has been removed from the table since is not significant. Liquid sample is withdrawn from the assay (Erlenmeyer Flask containing the liquid medium). This sample is passed through a filter.  As the assay continued and samples were taken, it was noted that there was a limited amount of liquid medium that could be taken from the flasks since liquid medium is not replaced in the assay, and a minimum liquid level must be maintained in the flask to keep the assay running. The sampling volume was then reduced to 50ml. This is insignificant since the DNA sample was then extracted from the filter and adjusted for metagenomic studies, and so having 70 ml or 50 ml of sample passed through the filter made no difference to the extracted samples.

If nine colonies were isolated, why just eight were analyzed for identification?:

Response: After revision, it was stated that this was a mistake. There are 7 isolates in Table 3, but it is specified as 8 in the text. Initially, 9 were isolated, but only 7 were identified. This has been corrected in the text.

In figure 1, indicate which DNA bands are expected or those of interest –

Response: Thanks. We have marked the DNA bands expected in red.

In Figure 3, it is not clear which microorganisms were used in the experiment –

Response: As described in the introduction and materials and methods section, leather biodegradation assay, according to ISO 20136, uses wastewater (it is possible to choose between municipal or tannery) as inoculum. In Table 2, samples M1, M2, and M3 are samples taken from municipal wastewater (M1), tannery wastewater (M2), and a mixture of both (50:50 ratio) (M3). The inoculum used for the assay is a mixture of both (M3). M3 is the initial inoculum for all assays.

The conclusions seens superficial. Conclusions do not mention the potential of selectively isolated bacteria, it is not clear how the identified microorganisms can have a dual capacity, leather biodegradation and resistance to heavy metals. 

Response: The rationale of the leather biodegradation assay is that the only carbon source for microorganisms is the tanned leather used in the assay. Therefore, the aim here is not only to study shifts in microbial diversity from one tanning agent to another but also to identify which species are increasing the most from initial inoculum since these would be the ones with the potential to degrade leather, which is tanned with metal-based tanning agents.

How the evaluation of the microbial communities present during the degradation process can improve leather biodegradation processes, what is the applicability of the data generated.

Response: These results could be used to improve and standardize the leather biodegradation assay since this methodology uses wastewater as inoculum. Each assay is irrepealable. These species could be used as a consortium to standardize the assay. On the other hand, the aim of further work from these results is to study the potential of a microbial consortium to bioagmentate leather composting processes, where species must be heavy metal resistant.

The manuscript has several technical/format issues to correct.

Response: The manuscript has been revised, and some typos have been corrected.

Reviewer 2 Report

Comments and Suggestions for Authors

The results of the comprehensive study of the microorganisms from tannery wastewaters for searching genera with biodegradation capacity are presented. The MS is relevant for the microbiology, the presented research is very important for the industry and nature protection. The reference list includes relevant publications on studied topic, included recent papers. The authors used proper experimental methodology: samples collection, species isolation, and molecular-genetic analysis. The results of the study are presented in the tables and high quality figures. The obtained sequenced were compared with the sequences from the public databases (NCBI and EZBioCloud). The conclusions of the MS are consistent and supported by data. The MS is well written and clear.

But authors should correct some errors before the publication.

Suggestions to authors:

Lines 43, 45, 102, 176 and further: Add the space before square brackets.

Lines 52, 80, 108, 127, 387 and further: Delete the empty lines.

Line 227: Correct ([48] to [48].

Line 297, 308:  I think, that using “percentages” instead “%” more correct.

Figure 4: Please, try to enlarge the legend for colors and their numbers.

Lines 393-398 and further: Specify the authors of the species and genus at the first mention.

Figure 5: Please, try to enlarge the numbers on OX, OY axes, and samples numbers.

Line 439: Brevundimonas should be written in italics.

Author Response

The results of the comprehensive study of the microorganisms from tannery wastewaters for searching genera with biodegradation capacity are presented. The MS is relevant for the microbiology, the presented research is very important for the industry and nature protection. The reference list includes relevant publications on studied topic, included recent papers. The authors used proper experimental methodology: samples collection, species isolation, and molecular-genetic analysis. The results of the study are presented in the tables and high-quality figures. The obtained sequenced were compared with the sequences from the public databases (NCBI and EZBioCloud). The conclusions of the MS are consistent and supported by data. The MS is well written and clear.

But authors should correct some errors before the publication.

Suggestions to authors:

Lines 43, 45, 102, 176 and further: Add the space before square brackets.

Response: Done

Lines 52, 80, 108, 127, 387 and further: Delete the empty lines.

Response:  Done

Line 227: Correct ([48] to [48].

Response: Done

Line 297, 308:  I think, that using “percentages” instead “%” more correct.

Response:  Done

Figure 4: Please, try to enlarge the legend for colors and their numbers:

Response: Done (Shown in Figure 4)

Lines 393-398 and further: Specify the authors of the species and genus at the first mention

Response: Done

Figure 5: Please, try to enlarge the numbers on OX, OY axes, and samples numbers.

Line 439: Brevundimonas should be written in italics:

Response: Done.

Reviewer 3 Report

Comments and Suggestions for Authors

Well presented.

1. Main question addressed is to explore the microbial diversity in leather biodegradation. The authors presented well dined data to address this question.

2.  The selective isolation and identification of the microbial species defined appear to be original.  I did not find any publication with this amount of detail in species identification.

3.  Biodegradability is a significant indicator of to evaluate the sustainability of leather. By elucidating the bacterial stains present in the wastewater samples, development of novel chrome-free tanning processes can be explored.

4.  Were there any statistical significances in figures 6-10 between species?

5. This paper accomplished its aim with the identification of tannery wastewaters.

6. As for the references, I did find a communication, Wang et al. Collagen and Leather (2024) 6:12 https://doi.org/10.1186/s42825-024-00151-z, that may be considered. Although the authors do have appropriate references on background information with regards to biodegradability of leather, I believe this communication is a recent overview.

7. The data presented are detailed and support the objective of this study.

Author Response

Comments and Suggestions for Authors

Well presented.

  1. Main question addressed is to explore the microbial diversity in leather biodegradation. The authors presented well dined data to address this question.
  2. The selective isolation and identification of the microbial species defined appear to be original.  I did not find any publication with this amount of detail in species identification.
  3. Biodegradability is a significant indicator of to evaluate the sustainability of leather. By elucidating the bacterial stains present in the wastewater samples, development of novel chrome-free tanning processes can be explored.

  1. Were there any statistical significances in figures 6-10 between species?

Response: No, no statistical significances were reported in figures 6-10 between species. The analysis primarily focuses on associating each sequence with an organism; grouping reads based on Family, Genus, or Species, and summarizing the presence of different taxonomic groups in the analyzed samples using tables and graphs. Additionally, a rarefaction curve analysis compares the number of analyzed sequences with the number of detected taxa at different levels, aiming to detect all organisms in the sample. If the curve reaches a saturation point, it indicates that all organisms have been detected. At the same time, an exponential phase suggests the need for more sequence data to capture the total variability of the sample.

  1. This paper accomplished its aim with the identification of tannery wastewaters.
  2. As for the references, I did find a communication, Wang et al. Collagen and Leather (2024) 6:12 https://doi.org/10.1186/s42825-024-00151-z, that may be considered. Although the authors do have appropriate references on background information with regards to biodegradability of leather, I believe this communication is a recent overview.

Response:  Thank you. We have added new paragraphs:

 More specifically, ISO 14851 is used to determine the biodegradability potential of a plastic material by measuring oxygen consumption during 2-6 months in a closed bottle set up where the degradation percentage is calculated as the ratio of oxygen consumption to the theoretical oxygen demand (ThOD)[38]. Lines 108-111

Recent studies have shown that chromium-free tanned leather has a higher degree of biodegradation and faster biodegradation rate than chrome-tanned leather. These have established that increasing tanning effects through enhancing tanning methods or conditions decreases leather biodegradability [38].

  1. The data presented are detailed and support the objective of this study.

Round 2

Reviewer 1 Report

Comments and Suggestions for Authors

The authors did not make a real effort to attend to the observations made in the previous version of the manuscript, and essentially, the manuscript still has the same concerns and issues to improve.

Commentaries

The description of the leather degradation experiments is difficult to understand. The treatments evaluated are presented mixed. Perhaps it would have been better to organize the data by type of treatment or leather sample.

The title of the manuscript does not really describe the experimental set or the results of the study. Some bacterial species were isolated, but no evidence of the relevance of such bacterial isolates for the leather biodegradation process was determined in the study. Moreover, heavy metal resistance was established, but no experiments for heavy metal resistance were carried out in the study.

In summary, the focus and description were not appropriate; the title does not reflect the main findings of the study. The biodegradation experiments, the most relevant in the study, are not adequately described, which makes it difficult to understand the results obtained.

The conclusions need to be adapted and delved into the relevant aspects of the study, and propose potential applications derived from its study to be implemented in improving leather treatments.

Additional commentaries

Line 25, use “seven” instead “7”

Lines 40, 42 and 43, review format of the Cr ions (Cr3+, Cr(III), and Cr(VI), choose one and be consistent in the whole manuscript

Line 43, complement “elevated pH” with the pH ranges

Lines 73-75, Review redaction, biodegradation does not apply to heavy metals

Line 106 and 111, add a space before references [37] and [38]

Line 144, in fragment “filtrate was supplemented with 0.5% yeast” it is yeast extract?, complement

Line 167, add a space before reference [41]

Lines 298-306, justify the text

In figure 3, it is not clear which microorganisms were used in the experiment

In figures 6b, 7b, 8b, 9b and 10b, present the data in figure caption in another way, is not easy to understand directly and make necessary to return to the table to find out the experimental condition.